# Wideband Beam Steering Concept for Terahertz Time-Domain Spectroscopy: Theoretical Considerations

**DOI:** 10.3390/s20195568

**Published:** 2020-09-28

**Authors:** Xuan Liu, Kevin Kolpatzeck, Lars Häring, Jan C. Balzer, Andreas Czylwik

**Affiliations:** Chair of Communication Systems (NTS), Faculty of Engineering, University of Duisburg-Essen (UDE), 47057 Duisburg, Germany; haering@nts.uni-duisburg-essen.de (L.H.); jan.balzer@uni-due.de (J.C.B.); Czylwik@nts.uni-duisburg-essen.de (A.C.)

**Keywords:** terahertz, photonic beam steering, time-domain spectroscopy, optical ring resonator, mode-locked laser diode

## Abstract

Photonic true time delay beam steering on the transmitter side of terahertz time-domain spectroscopy (THz TDS) systems requires many wideband variable optical delay elements and an array of coherently driven emitters operating over a huge bandwidth. We propose driving the THz TDS system with a monolithic mode-locked laser diode (MLLD). This allows us to use integrated optical ring resonators (ORRs) whose periodic group delay spectra are aligned with the spectrum of the MLLD as variable optical delay elements. We show by simulation that a tuning range equal to one round-trip time of the MLLD is sufficient for beam steering to any elevation angle and that the loss introduced by the ORR is less than 0.1 dB. We find that the free spectral ranges (FSRs) of the ORR and the MLLD need to be matched to 0.01% so that the pulse is not significantly broadened by third-order dispersion. Furthermore, the MLLD needs to be frequency-stabilized to about 100 MHz to prevent significant phase errors in the terahertz signal. We compare different element distributions for the array and show that a distribution according to a Golomb ruler offers both reasonable directivity and no grating lobes from 50 GHz to 1 THz.

## 1. Introduction

Wideband terahertz systems, operating in the frequency range between the microwave and far-infrared region of the electromagnetic spectrum, have the potential to combine the best of two worlds. On the one hand, they offer hundreds of GHz of spectral bandwidth and sub-mm spatial resolution. On the other hand, they have the capability to see through dielectric materials that are opaque to infrared and visible light. These properties make them attractive for a variety of applications, including material identification [1,2], material characterization [3,4], and imaging [5,6]. However, while electronic terahertz systems already exhibit a high degree of integration [7,8], their instantaneous bandwidths have yet failed to exceed a few tens of GHz. Photonic, i.e., laser-driven, terahertz time-domain spectroscopy (THz TDS) systems on the other hand can already provide a few THz of bandwidth at the cost of a low degree of integration and high system complexity. These systems use trains of ultra-short infrared pulses that are photodetected to generate broadband terahertz pulse trains. On the receiver side, the terahertz pulses are sampled with a time-delayed version of the same infrared pulse train [3]. State-of-the-art fiber-coupled THz TDS systems employ mode-locked fiber lasers at 1550 nm with repetition rates around 100 MHz as their driving light source. These systems exhibit bandwidths of several THz and can be shrunk to shoe box size [9]. However, a remaining challenge is the limited range, which decreases drastically with increasing frequency due to the low terahertz transmit power [10,11]. Furthermore, a decrease in size is not possible because the fiber laser inhibits integration. While state-of-the-art THz TDS systems already provide acquisition rates of a few kHz, spatially resolved measurements are still slow due to the need for a mechanical translation or rotation of either the object under test or the emitter and detector. We aim to increase the range of THz TDS systems and enable high-speed non-mechanical steering of the emitted terahertz beam by using a linear array of *N* coherently driven spatially distributed emitters instead of a single emitter. This way, both the directivity and—through spatial power combining—the total radiated power can be increased, yielding a possible increase of the equivalent isotropically radiated power by $N^{2}$. The emitted beam can be steered by adjusting the time delays between the optical signals at the photonic terahertz emitters. This concept is known from photonic communications and is commonly referred to as photonic true time delay beam steering [12,13,14,15,16]. The delays introduced in the infrared domain are translated directly to the terahertz domain. In the terahertz range, it has been successfully demonstrated by Bauerschmidt et al. for frequency-domain, i.e., continuous-wave, spectroscopy using individually lensed fiber-coupled terahertz emitters and fiber-coupled free-space variable optical delay lines [17]. In a more recent approach, Preu et al. have replaced the discrete emitters with a monolithically integrated array of antenna-integrated photodiodes [18]. However, the fiber-coupled free-space variable optical delay lines are still bulky and mechanically sensitive. Likewise, a fiber-coupled beam steering network is prohibitively complex even for a modest number of array elements. The solution is the use of integrated optics for the photonic beam steering network. Most prominently, integrated optical ring resonators (ORRs) have been proposed by Meijerink et al. [14] and successfully implemented by Zhuang et al. [15] as variable optical delay units (ODUs) for X-band communications. More recently, ORRs have been demonstrated for photonic beam steering for W-band communications by Liu et al. [16]. The group delay response of ORRs is intrinsically narrowband and periodic in the frequency domain. Although bandwidths of a few GHz have been reported by cascading several rings [16], this is still orders of magnitude too small for THz TDS.

In this paper, we propose a concept that exploits the periodicity of the ORRs’ group delay response to circumvent their bandwidth limitation. We choose the optical perimeter of the ORRs so that their free spectral range (FSR) is identical to the FSR of the mode-locked laser that drives the THz TDS system. This way the laser modes can be aligned with the resonant frequencies of the ORR. Consequently, the ORR introduces identical delays to all laser modes. To keep the ORRs small enough to be realized in an integrated optical circuit, the FSR needs to be on the order of several tens of GHz. This condition is fulfilled by using a monolithic mode-locked laser diode (MLLD) in an approach we call ultra-high repetition rate terahertz time-domain spectroscopy (UHRR THz TDS). This approach has first been demonstrated by Merghem et al. [19] and system-theoretically analyzed by the authors [20]. Besides its suitability for photonic beam steering, UHRR THz TDS has the advantage that the light source has the potential for monolithic integration, is electrically pumped, supplies an optical power of around 100 mW without amplification, and operates at 1550 nm telecom wavelengths [21,22]. Another practical advantage of the ultra-high repetition rate is the fact that identical pulses are detected every few picoseconds. This alleviates the need for accurate matching of the transmit and receive path lengths and allows the use of a much shorter ODU in the receiver arm. Particularly in reflection mode, where the distance to the target may not be known, this is highly beneficial. The disadvantage of UHRR THz TDS is its comparatively low bandwidth of typically less than 2 THz and poor spectral resolution of a few tens of GHz [19,20,23]. The bandwidth is primarily determined by the optical bandwidth of the MLLD and the low-pass behavior of the terahertz components. The spectral resolution is equal to the repetition rate of the MLLD.

The paper is structured as follows. In Section 2, we thoroughly analyze, both analytically and numerically, the ORR as a variable delay element for frequency-discrete optical signals. We describe its principle of operation and discuss possible implementations. We deduct the relationship between the optical group delay of the ORR and the delay of the resulting terahertz signal. Finally, we investigate the effects of non-idealities and give estimates for the required tolerances and stability.

In Section 3, we address the challenge of operating an antenna array across the entire bandwidth of the UHRR THz TDS system. We show that for frequency-discrete terahertz signals a variable delay range equal to one round-trip time of the MLLD is sufficient for steering the beam to any elevation angle. In this *quasi*-true time delay approach the direction of the main beam is frequency-independent. Furthermore, we consider the design for wideband operation of a linear array and compare different element distributions for the frequency range from 50 GHz to 1 THz.

## 2. Integrated Optical Ring Resonator-Based ODUs for MLLDs

ORRs play an important role in integrated photonic circuits as components with multiple functionalities. They have been used as optical switches [24], filters [25], modulators [26], and ODUs in photonic beam steering networks [14,15,16]. For applications that require wideband operation, ORRs with slightly different resonant frequencies have been cascaded to achieve a flat group delay response within a couple of GHz [16]. The ORR has a group delay characteristic that is periodic in frequency with an FSR that is determined by the optical perimeter of the ring. Together with a MLLD whose modes are also periodically spaced, this periodic group delay characteristic offers a true time delay solution with a bandwidth of a couple of terahertz. We design the ring so that the FSRs of the ring resonator and the MLLD are identical, thus the frequencies of the laser modes can be aligned with the resonant frequencies of the ORR. Consequently, the ORR introduces identical delays to all laser modes. After photomixing, the same delay is then translated to the resulting terahertz signal. The delay that is introduced by the ring resonator is adjusted by varying the coupling coefficient between the ring and the bus.

One of the means for adjusting the coupling coefficient is using a Mach–Zehnder modulator as shown in Figure 1. The Mach–Zehnder modulator consists of two 3 dB couplers and a phase shifter. One of the outputs of the modulator is fed back to one of the inputs of the modulator, thus forming the ring. By varying the phase difference Δψ between the two arms which connect the two couplers, the coupling coefficient between the bus and ring can be tuned. Consequently, the delay introduced by the ring resonator is adjusted. The resonant frequencies of the ORR can be changed by adjusting the round-trip phase shift Δϕ while other characteristics of the ORR remain the same.

In Section 2.1, we analytically calculate the transfer function of the Mach–Zehnder modulator-based ORR. The group delay response and transmission spectrum of the ORR are then numerically evaluated. Based on the calculated transfer function of the ORR, we investigate the effects of the inherent frequency dependence of the waveguide-based phase shifters, FSR mismatch, and fluctuations of the laser frequency on the resulting terahertz pulse in Section 2.2, Section 2.3, Section 2.4, respectively.

### 2.1. Mathematical Model of a Mach–Zehnder Modulator-Based ORR

To obtain the group delay response and the transmission spectrum of the ring resonator, we calculate its transfer function. As shown in Figure 2, we introduce the wave amplitudes $a_{c1,1}$ and $a_{c1,2}$ as the incoming wave amplitudes on the left-hand side of coupler 1, and $b_{c1,3}$ and $b_{c1,4}$ as the outgoing wave amplitudes on the right-hand side of this coupler. Likewise, the variables $a_{c2,1}$ and $a_{c2,2}$ denote the incoming wave amplitudes on the left-hand side of coupler 2, and $b_{c2,3}$ and $b_{c2,4}$ denote the outgoing wave amplitudes on the right-hand side.

We assume that the couplers are reflection-free and lossless and the length of two couplers together with the arms in between can be neglected compared to the perimeter of the ring. Furthermore, the frequency dependence of the phase shifts Δψ and Δϕ is neglected for this subsection. It will be addressed in detail in Section 2.2. In terms of the incoming and outgoing waves we have that
(1)$$\begin{array}{cc} \left(\begin{array}{c} b_{c1,3}\\ b_{c1,4} \end{array}\right) & =\left(\begin{array}{cc} \frac{1}{\sqrt{2}} & j\frac{1}{\sqrt{2}}\\ j\frac{1}{\sqrt{2}} & \frac{1}{\sqrt{2}} \end{array}\right)\cdot \left(\begin{array}{c} a_{c1,1}\\ a_{c1,2} \end{array}\right), \end{array}$$
(2)$$\begin{array}{cc} \left(\begin{array}{c} a_{c2,1}\\ a_{c2,2} \end{array}\right) & =\left(\begin{array}{cc} e^{-j\Delta \psi } & 0\\ 0 & 1 \end{array}\right)\cdot \left(\begin{array}{c} b_{c1,3}\\ b_{c1,4} \end{array}\right), \end{array}$$
(3)$$\begin{array}{cc} \left(\begin{array}{c} b_{c2,3}\\ b_{c2,4} \end{array}\right) & =\left(\begin{array}{cc} \frac{1}{\sqrt{2}} & j\frac{1}{\sqrt{2}}\\ j\frac{1}{\sqrt{2}} & \frac{1}{\sqrt{2}} \end{array}\right)\cdot \left(\begin{array}{c} a_{c2,1}\\ a_{c2,2} \end{array}\right),\;\mathrm{and}\; \end{array}$$
(4)$$\begin{array}{cc} a_{c1,1} & =e^{-\left[\alpha \left(\omega \right)+j\beta \left(\omega \right)\right]\cdot L}\cdot e^{-j\Delta \phi }\cdot b_{c2,3}, \end{array}$$
where *L* is the perimeter of the ring, $\alpha \left(\omega \right)$ is the attenuation constant of the waveguide, and $\beta \left(\omega \right)$ is the phase constant of the waveguide. Substituting Equations (Equation 1), (Equation 2) and (Equation 4) into Equation (Equation 3) gives the transfer function of the Mach–Zehnder-based ORR
(5)$$H\left(\omega ,\Delta \psi ,\Delta \phi \right)=\frac{b_{c2,4}}{a_{c1,2}}=\frac{{\left(\frac{j}{2}\cdot e^{-j\Delta \psi }+\frac{j}{2}\right)}^{2}\cdot e^{-\left[\alpha \left(\omega \right)+j\beta \left(\omega \right)\right]\cdot L}\cdot e^{-j\Delta \phi }}{1-\left(\frac{1}{2}\cdot e^{-j\Delta \psi }-\frac{1}{2}\right)\cdot e^{-\left[\alpha \left(\omega \right)+j\beta \left(\omega \right)\right]\cdot L}\cdot e^{-j\Delta \phi }}+\left(\frac{1}{2}-\frac{1}{2}\cdot e^{-j\Delta \psi }\right).$$

The perimeter *L* of the ring is determined as
(6)$$L=\frac{c_{0}}{n_{g}\cdot f_{\mathrm{FSR},\mathrm{ORR}}},$$
where $c_{0}$ is the speed of light in vacuum and $n_{g}$ is the group index of the waveguide. With the FSR $f_{\mathrm{FSR},\mathrm{ORR}}$ of the ORR equal to the FSR of the MLLD $f_{\mathrm{FSR},\mathrm{MLLD}}$, each mode of the MLLD experiences the same delay. For the following numerical analysis, we consider a MLLD with the FSR $f_{\mathrm{FSR},\mathrm{MLLD}}=50\;\mathrm{GHz}$ which has a spectrum that is centered at 193.41 THz. We consider here the TriPleX waveguide, which is made of a ${\mathrm{SiO}}_{2}$ cladding and a ${\mathrm{Si}}_{3}N_{4}$-${\mathrm{SiO}}_{2}$ core for the following numerical analyses. TriPleX is a commercially accessible technology from the company LioniX. Among various TriPleX waveguide geometries, we choose the symmetric double strip due to its small minimum bend radius and low waveguide propagation loss [27]. Its properties are listed below.

Attenuation constant α=0.1dB/cm.Effective refractive index $n_{\mathrm{eff}}=1.535$ at the wavelength $\lambda_{0}=1550\;\mathrm{nm}$.Group index $n_{g}=1.72$.

With the information of the effective refractive index $n_{\mathrm{eff}}$ and the group index $n_{g}$ of the waveguide, the wavelength-dependent effective refractive index of the waveguide can be calculated. Therefore, the wavelength-dependent phase constant $\beta \left(\lambda \right)$ can be calculated as
(7)$$\begin{array}{c} \beta \left(\lambda \right)=\frac{2\pi }{\lambda }\cdot n_{\mathrm{eff}}\left(\lambda \right)=\frac{2\pi }{\lambda }\cdot \left[n_{\mathrm{eff}}-\left(\frac{n_{\mathrm{eff}}-n_{g}}{\lambda_{0}}\cdot \left(\lambda -\lambda_{0}\right)\right)\right], \end{array}$$
where λ is the wavelength in vacuum. The group delay response of the Mach–Zehnder-based ring resonator is then calculated as
(8)$$\tau_{\mathrm{gr}}=-\frac{\partial ∠\left(H\left(\omega ,\Delta \psi ,\Delta \phi =0^{\circ }\right)\right)}{\partial \omega },$$
and shown for different Δψ in Figure 3. With no phase differences between the arms $\left(\Delta \psi =0\right)$, the group delay shows a flat response over the entire frequency range. In this case, every laser mode experiences the same minimum delay which is exactly one round-trip time of the ORR. According to the quasi-true time delay concept, which will be proposed in Section 3, a MLLD with the FSR $f_{\mathrm{FSR},\mathrm{MLLD}}=50\;\mathrm{GHz}$ requires a delay range of 20 ps to allow beam steering to any elevation angle. Therefore, the ORR must have a tuning range up to 40 ps, which can be easily achieved by adjusting the phase difference between the arms as depicted in Figure 3 for $0\leq \Delta \psi <39^{\circ }$.

In Figure 3, we observe that with increasing group delay $\tau_{\mathrm{gr}}$, the attenuation at the resonant frequencies and the quality factor of the ORR increase. As the proposed quasi-true time delay concept limits the maximum required delay to one round-trip time of the MLLD cavity, the maximum attenuation is restricted to less than 0.1 dB. Moreover, we observe that with increasing group delay $\tau_{\mathrm{gr}}$, the resonant frequencies of the ORR shift from the desired resonant frequencies. The round-trip phase Δϕ therefore needs to be adjusted to align the resonant frequencies while the other characteristics of the group delay response remain the same as shown in Figure 4.

### 2.2. Dispersion Due to the Phase Shifter

For simplicity, the characteristics of the ring resonator shown above are calculated by Equation (Equation 5) in which the frequency dependence of the phase shifts Δψ and Δϕ is neglected. However, the phase response of waveguide-based phase shifters is intrinsically frequency dependent regardless of their implementation. In this section, we consider the phase shifters to be thermo-optic phase shifters to investigate the effect of the frequency dependence on the group delay response of the ORR.

For a thermo-optic phase shifter, the change of phase $\Delta \Phi_{0}$ as a function of the change of temperature ΔT at the frequency $f_{0}$ is
(9)$$\Delta \Phi_{0}=\Delta \Phi \left(f=f_{0},\Delta T\right)=\frac{2\pi L_{h}\cdot f_{0}}{c_{0}}\cdot \frac{\partial n}{\partial T}\cdot \Delta T,$$
where $L_{h}$ is the length of the microheater and $\frac{\partial n}{\partial T}$ is the thermo-optic coefficient of the waveguide. The relative change of phase as a function of the change of frequency Δf is
(10)$$\frac{\Delta \Phi \left(f=f_{0}+\Delta f,\Delta T\right)-\Delta \Phi_{0}}{\Delta \Phi_{0}}=\frac{\Delta f}{f_{0}}.$$

The frequency-dependent phase response of the thermo-optic phase shift is then
(11)$$\Delta \Phi \left(f=f_{0}+\Delta f\right)=\frac{f_{0}+\Delta f}{f_{0}}\cdot \Delta \Phi_{0}.$$

Taking into account this frequency-dependent phase response of the phase shifters in Equation (Equation 5), the delays τ that the individual laser modes experience are calculated and plotted in Figure 5a. The delay exhibits a linear increase with frequency. The laser pulse is thus linearly chirped. To observe the effects on the emitted terahertz pulse, we first establish the mathematical relationship between the complex optical spectrum incident on the photomixer and the emitted terahertz spectrum. The terahertz field in the far-field region of the antenna is proportional to the first time derivative of the incident instantaneous optical power $p_{\mathrm{opt}}\left(t\right)$ incident on the photomixer. Therefore, the resulting terahertz spectrum can be estimated by the instantaneous optical power, and the delay of the terahertz pulse is identical to the delay of the laser pulse. To simplify the analysis, we neglect the frequency dependence of the photomixer and the antenna, and let the instantaneous optical power $p_{\mathrm{opt}}\left(t\right)$ represent the emitted terahertz signal. It can be calculated as [20,28]
(12)$$\begin{array}{c} p_{\mathrm{opt}}\left(t\right)\propto \sum_{k=0}^{N-1}E_{k}^{2}\;+\;2\cdot \sum_{m=1}^{N-1}\sum_{k=m}^{N-1}E_{k}E_{k-m}\cdot \cos\left(2\pi \cdot m\cdot f_{\mathrm{FSR},\mathrm{MLLD}}\cdot t+\varphi_{k}-\varphi_{k-m}\right), \end{array}$$
where $E_{k}$ and $\varphi_{k}$ are the amplitude and phase of the *k*-th mode of the MLLD, respectively, and *N* is the number of significant laser modes. In the following, we consider a laser with 40 significant modes and a rectangular spectrum, i.e., N=40 and $E_{k}=E_{0}\;\forall \;k$. As depicted in Figure 5b, the chromatic dispersion does not significantly broaden the resulting terahertz pulse. However, the decrease of the pulse maximum with increasing delay indicates a worsening in the photomixing efficiency. Furthermore, the ring resonator introduces a different delay to the terahertz pulse than to the individual laser modes. For example, a chirped delay of around 25 ps of the laser modes results in a delay of 30.67 ps to the terahertz pulse (cf. Figure 5b). By calculating the first derivative of the delay that the laser modes experience in Figure 5a with respect to frequency, we confirm that the delay increases linearly with frequency. For a linearly chirped laser pulse, its delay can be derived as
(13)$$\tau_{\mathrm{THz}}=\tau_{0}+f_{0}\cdot \frac{\partial \tau \left(\tau_{0},f\right)}{\partial f}+f_{\mathrm{FSR}}\cdot \left(N-1\right)\cdot \frac{\partial \tau \left(\tau_{0},f\right)}{\partial f}\;.$$

With $\tau_{0}$ as the delay of the first significant mode of the MLLD at the frequency $f_{0}$, the calculated slope $\frac{\partial \tau \left(\tau_{0},f\right)}{\partial f}$ and the calculated delays $\tau_{\mathrm{THz}}\left(\tau_{0}\right)$ of the terahertz pulse are depicted in Figure 6. It shows that the ring resonator needs to introduce less delay to the linearly chirped laser pulse than the non-chirped one for the same delay of the resulting terahertz pulse. For the required 40 ps delay of the terahertz pulse, an ORR that introduces a delay around 30 ps is sufficient. As mentioned in Section 2.1, the shorter the delay is, the less the attenuation of the ORR is. This brings an unexpected advantage in terms of the optical power of the laser pulse.

### 2.3. Dispersion Due to FSR Mismatch

The crucial requirement of the proposed concept for introducing identical delays to all laser modes is perfect alignment between the modes of the MLLD and the resonances of the ORR. However, the FSRs of the ORR and the MLLD may be slightly different due to the fabrication inaccuracy. In this section, we investigate the effect on the resulting terahertz pulse for different FSR mismatches.

For illustrative purposes, a case where the FSRs of the ORR and the MLLD are severely mismatched (i.e., a deviation of 1%) is depicted in Figure 7. Here, we align the frequency of the first significant mode of the MLLD with one of the resonances of the ORR. The FSR mismatch causes the laser modes to slip out of the resonances of the ORR whose group delay response is bell-shaped. Thus, a nonlinear chirp is introduced to the optical pulse and hence to the terahertz pulse.

With increasing group delay the bandwidth of the ORR decreases. As a consequence, the laser chirp due to the FSR mismatch becomes more severe. To consider the worst-case scenario, we thus investigate an ORR with a group delay around 30 ps. Figure 8a shows the delays that ORRs with different FSR mismatches introduce to the laser modes and Figure 8b shows the resulting terahertz pulses. Up to an FSR mismatch of 0.01%, the resulting terahertz pulse is not significantly broadened. However, the effect of the FSR mismatch can be observed by a slight decrease of the terahertz pulse amplitude. For an FSR mismatch above 0.01%, the ORR introduces an evident nonlinear chirp. This nonlinear chirp results in a severe distortion of the terahertz pulse. This causes the bandwidth and power of the resulting terahertz pulse to be significantly reduced.

To quantify the nonlinear chirp and the effect on the terahertz pulse, we show the third order dispersion $\frac{\partial^{2}\tau }{\partial f^{2}}$ and the resulting terahertz pulse width (full-width at half-maximum (FWHM)) as a function of the FSR mismatch in Figure 9. The plot shows that the pulse width is virtually constant up to a mismatch of 0.01% and drastically increases above that value.

### 2.4. Delay Fluctuation Due to Frequency Instability

The instabilities of the injection current and the temperature control of the MLLD can lead to fluctuations of its center frequency and FSR. Consequently, the laser modes drift around the resonances of the ORR. Under stable mode-locked conditions, the FSR of the MLLD changes less than 1 MHz, whereas the fluctuation of the center frequency can be tens of MHz. Therefore, we consider the FSR of the MLLD to be constant. Figure 10 shows the change of the delay as a function of the change of the center frequency. At $\tau_{0}=20\;\mathrm{ps}$, the ORR has a flat group delay over the entire frequency range, as shown in Figure 3 and Figure 4. Therefore, the center frequency fluctuations of the MLLD cause no delay fluctuations. Phase stability of 1${}^{\circ }$ up to 2 THz requires any delay fluctuations to be less than 1.4 fs. Thus, in the worst case scenario where the ring resonator has a delay of 30 ps at the resonances of the ORR, the center frequency of the MLLD has to be stabilized within 97 MHz. This can be easily achieved by a high precision temperature controller and current source.

## 3. Photonic Quasi-True Time Delay Beam Steering for UHRR THz TDS

In order to perform photonic beam steering on the transmitter side of a wideband UHRR THz TDS system, both the photonic beam steering network and the antenna array need to operate across a large bandwidth. While the ODUs need to introduce the same delay across the entire optical spectrum, the antenna array needs to be designed and operated in a way that the main beam direction is independent of frequency. At the same time, the radiation pattern needs to exhibit high directivity and no grating lobes across the entire range of emitted terahertz frequencies. The first challenge has been addressed in Section 2. The second challenge will be addressed in this section. In Section 3.1, we explain the design and the operation of the photonic beam steering network. We give an expression for the array factor for arbitrary element distributions and determine the delays that are required to steer the beam in a desired direction. In Section 3.2, we highlight some of the design considerations for the antenna array and the trade-offs that need to be made for wideband operation. We propose three different element distributions with vastly different characteristics and analyze these with respect to directivity, beamwidth, grating lobes, and sidelobe level. Although receiver-side beam steering is outside the scope of this paper, it should be noted that these investigations are equally applicable for an array of detectors.

### 3.1. Design and Operation of the Quasi-True Time Delay Beam Steering Network

We consider a linear array of identical antenna-integrated photodiodes that are fed through a photonic beam steering network. For an *N*-element array of antennas arranged along the *z*-axis, as depicted in Figure 11, the optical signal is split into *N* branches and the signal in each branch is delayed by the delay $\tau_{n}$.

The delayed signals are fed into antenna-integrated photodiodes to generate the terahertz photocurrents $I_{\mathrm{THz},n}\left(\omega \right)$ through photomixing. The photocurrents excite the antennas of the array to emit terahertz radiation. As described in Section 2, the electrical field that is generated in the far field of each antenna is proportional to the first time derivative of the instantaneous optical power $p_{\mathrm{opt},n}\left(t\right)$ at the respective photodiode. Thus, neglecting any dispersive effects in the photonic beam steering network, the time delay introduced in the optical domain is directly translated to the radiated electrical fields.

If z=0 is taken as the phase center of the array, the spectral component at the frequency $f=m\cdot f_{\mathrm{FSR},\mathrm{MLLD}}$ of the electric field radiated by the *n*-th element has the phase lead
(14)$$\begin{array}{c} \psi_{n}\left(\theta ,f=m\cdot f_{\mathrm{FSR},\mathrm{MLLD}}\right)=2\pi \cdot m\cdot f_{\mathrm{FSR},\mathrm{MLLD}}\cdot \left(\frac{z_{n}\cdot \cos\theta }{c_{0}}-\tau_{n}\right) \end{array}$$
at the elevation angle θ in the far field of the array, where $c_{0}$ is the speed of light in air. By superimposing the contributions from all *N* elements, the array factor results [29]
(15)$$\begin{array}{c} g\left(\theta ,f=m\cdot f_{\mathrm{FSR},\mathrm{MLLD}}\right)=\sum_{n=1}^{N}e^{j\psi_{n}\left(\theta ,f=m\cdot f_{\mathrm{FSR},\mathrm{MLLD}}\right)} \end{array}$$

Because the emitted terahertz pulse repeats itself once every $f_{\mathrm{FSR},\mathrm{MLLD}}^{-1}$, the required delay range of the variable optical delay elements is one round-trip time of the laser cavity independent of the size of the array and the elevation angle to which the beam is steered. A radiation maximum in the direction $\theta =\theta_{0}$ is achieved at any frequency $f=m\cdot f_{\mathrm{FSR},\mathrm{MLLD}}$ by choosing the delays
(16)$$\begin{array}{c} \tau_{n}\left(\theta_{0}\right)=\frac{z_{n}\cdot \cos\theta_{0}}{c_{0}}\;\mathrm{mod}\;\frac{1}{f_{\mathrm{FSR},\mathrm{MLLD}}}\;. \end{array}$$

We call this concept photonic *quasi*-true time delay beam steering. As has been shown in Section 2, the small required delay range is highly beneficial when the delay elements are realized as ORRs because it allows the use of relatively low-quality factor resonators. It should be noted that the delays introduced to the emitted terahertz signals for beam steering will alter the spectral phase of the detected terahertz signal in the UHRR THz TDS system. However, the spectral phase can be equalized by calibration against a reference measurement.

### 3.2. Design Considerations for the Antenna Array

The design of a linear array allows as main design parameters the number of elements *N*, the length of the array *L*, the spatial distribution of the elements along the length of the array, the amplitude distribution, and the element pattern of the antennas. A good array design aims to fulfill the following criteria across the usable frequency range.

The directivity of the main lobe should be as high as possible.Side-lobes should be suppressed as much as possible and grating lobes should not occur.The beam should be steerable across the entire elevation range $\theta_{0}=0\;\ldots \;180^{\circ }$.

Although the usable frequency range, e.g., f=50GHz…1THz, is less than the optical bandwidth of the MLLD due to the low-pass behavior of the antenna-integrated photodiodes, it still spans several octaves. Thus, there is obviously a compromise between some of these design goals. Most notably, the behavior of the array will inevitably change across the extremely large targeted frequency range. Because the number of elements largely determines the technological effort and system complexity, it is reasonable to choose *N* as a constraint and allow the length of the array as well as the distribution of the elements to be arbitrarily chosen. Furthermore, as transmit power is at a premium at terahertz frequencies, it is advisable to choose directivity over sidelobe level and thus use a uniform amplitude distribution, i.e., equal amplitudes for all elements of the array. For reasons of simplicity, we consider isotropic antennas. This leaves the element distribution and the length of the array as the design parameters. For our analysis we choose a 10-element array, so that each photodiode receives sufficient optical power from the MLLD, which has an output power of roughly 100 mW. We propose the following three element distributions as depicted in Figure 12 and analyze these in the frequency range f=50GHz…1THz with $f_{\mathrm{FSR},\mathrm{MLLD}}=50\;\mathrm{GHz}$ with respect to the design criteria given above:Dense uniform distribution with $z_{n}=n\cdot d_{\mathrm{dense}}$ with $d_{\mathrm{dense}}=\frac{c_{0}}{2\cdot 1\;\mathrm{THz}}$.Sparse uniform distribution with $z_{n}=n\cdot d_{\mathrm{sparse}}$ with $d_{\mathrm{sparse}}=\frac{c_{0}}{2\cdot 50\;\mathrm{GHz}}$.Nonuniform distribution according to a 10-mark Golomb ruler with $z_{n}=\left\{0,\;1,\;6,\;10,\;23,\;26,\;34,\;41,\;53,\;55\right\}\cdot d_{\mathrm{dense}}$.

The two uniform distributions represent two extreme cases. In the case of the dense distribution, the array is optimized for the highest occurring frequency, whereas in the case of the sparse distribution, the array is optimized for the lowest occurring frequency. The former case will lead to low directivity and a wide beamwidth at lower frequencies, whereas the latter case will lead to the appearance of grating lobes for f≥100GHz. Golomb rulers have previously been proposed for zero-redundancy arrays [30,31,32]. These allow the construction of large and highly directive arrays with few elements. While the focus in these papers is maximum possible angular selectivity for a given number of array elements, another paper has discussed sparse arrays with optimized element distributions for wideband applications [33]. We expect the zero-redundancy characteristic of the Golomb ruler to provide good performance across our extremely large targeted frequency range.

Calculated frequency-dependent radiation patterns for all three distributions are depicted in Figure 13 for the broadside case $\theta_{0}=90^{\circ }$. The elevation angle is denoted on the horizontal axis, the frequency is given on the vertical axis, and the resulting directivity is color-coded. A directivity of 10 dBi is depicted in dark red and a directivity less or equal to -10 dBi is depicted in dark blue. In the case of a dense element spacing (Figure 13a), radiation is almost isotropic at the lower end of the spectrum and increases in directivity with increasing frequency. At the same time, the beam narrows towards higher frequencies. The plot for the sparse array (Figure 13b) exhibits a narrow beam with an increasing number of grating lobes. The beam narrows with increasing frequency. Because the directivity is equally “shared” between the main lobe and the grating lobes, the directivity is constant and equal to the number of elements D=N for all frequencies $f=m\cdot f_{\mathrm{FSR},\mathrm{MLLD}}$. As expected of a uniform array, we find that the sidelobe level is constant around 13 dB.

The array with an element distribution according to the Golomb ruler (Figure 13c) is well-behaved across the entire frequency range. It can be found that a much narrower beam can be realized across the entire frequency range than with the dense uniform array without suffering from the appearance of grating lobes as the sparse uniform array does. However, this distribution exhibits much higher sidelobes than the uniform arrays. The sidelobe level for the Golomb ruler distribution is independent of the steering angle. It is about 10.3 dBi at 50 GHz and constant at around 7.1 dBi from 100 GHz to 1 THz. These results show that nonuniform arrays trade the suppression of grating lobes for a higher sidelobe level. Due to practically unavoidable tolerances of array geometries and coupling between the antennas, radiation patterns will practically be slightly different. The consideration of deviating antenna patterns goes beyond the scope of this paper and may be analyzed in future work.

For a quantitative comparison, the direction-dependent directivity and beamwidth are numerically evaluated at several discrete frequencies as shown in Figure 14. Most notably, the narrowest beamwidths and a directivity that is independent of both frequency and main beam direction are achieved for the sparse uniform array, whereas for the dense array the directivity increases with frequency and the beam widens towards the end-fire positions. However, these plots conceal the detrimental effects of grating lobes with regard to angular ambiguity. The dense uniform array is a reasonable choice in applications where the objective of the array is to increase the range without a need for angular selectivity. In UHRR THz TDS the transmit power decreases drastically with frequency. Thus, a dense array, where the directivity increases with frequency, has the potential to equalize the received signal.

The directivity in the case of the Golomb ruler is significantly higher at low frequencies than that of a dense uniform array and reaches its maximum of 10 dBi at 1 THz. The beamwidth is close to that of the sparse uniform array around broadside. However, the beam widens more significantly towards the end-fire directions. The Golomb ruler is an excellent choice for applications that require a reasonable angular selectivity across the entire frequency range.

## 4. Conclusions and Future Perspectives

In this work, we have proposed a solution for beam steering on the transmitter side of wideband UHRR THz TDS systems. Our quasi-true time delay beam steering concept is based on integrated ORRs as ODUs in a photonic network driving a linear array of antenna-integrated photodiodes. The amplitude and phase information of the UHRR THz TDS system can be recovered by calibration against a reference measurement. The intrinsic bandwidth limitation of the ORRs is overcome by aligning their resonances with the modes of the MLLD that drives the UHRR THz TDS system. We have studied the feasibility of the ORR-based ODUs analytically and numerically. Our analysis shows that the intrinsic dispersion of the integrated optical phase shifters leads to a small chirp of the optical spectrum. This chirp causes the delay of the generated terahertz signal to differ from the group delay of the optical signal in a well-defined way. We have paid particular attention to the effects of a potential mismatch between the FSRs of the MLLD and the ORR. Numerical analysis shows that they need to match to within 0.01% so as not to severely affect the generated terahertz pulse and that the laser needs to be stabilized within 100 MHz for the phase stability to be better than 1${}^{\circ }$ up to 2 THz. We have shown that a delay range equal to one round-trip time of the MLLD is sufficient for the ODUs to enable beam steering to any elevation angle with a frequency-independent main beam direction. An element distribution according to a Golomb ruler is particularly advantageous in applications that require reasonably high directivity and good angular selectivity across the entire frequency range without the occurrence of grating lobes. We will present the experimental demonstration of our wideband beam steering concept in a future second part of this work. Therefore, a prototype of the ORR will be fabricated on the TriPleX platform by a commercially available foundry process and thoroughly characterized.

## Figures and Tables

**Figure 1 sensors-20-05568-f001:**
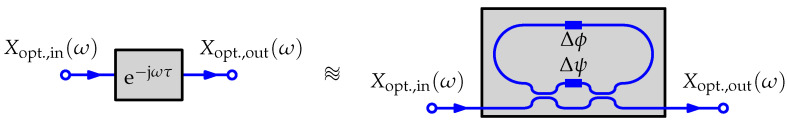
Mach–Zehnder optical ring resonator (ORR) as the optical delay unit (ODU) for photonic true time delay beam steering.

**Figure 2 sensors-20-05568-f002:**
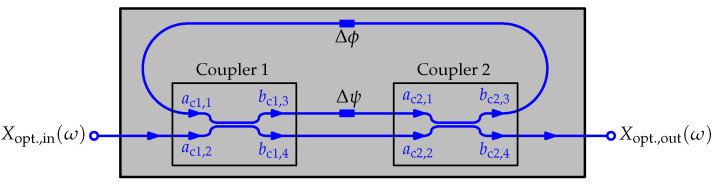
Signal flow graph representation of a Mach–Zehnder-based ORR.

**Figure 3 sensors-20-05568-f003:**
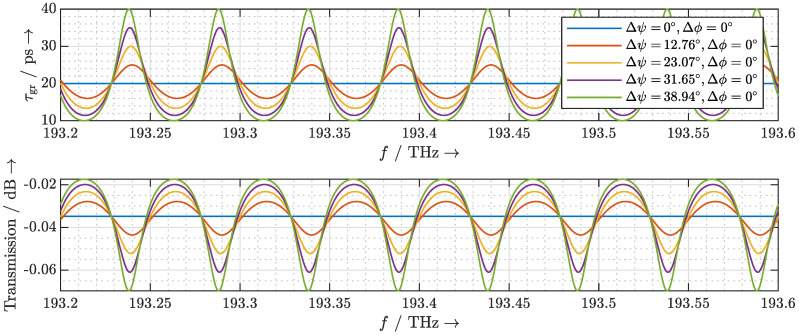
Calculated group delay response and transmission spectrum of an ORR with $f_{\mathrm{FSR},\mathrm{ORR}}=50\;\mathrm{GHz}$.

**Figure 4 sensors-20-05568-f004:**
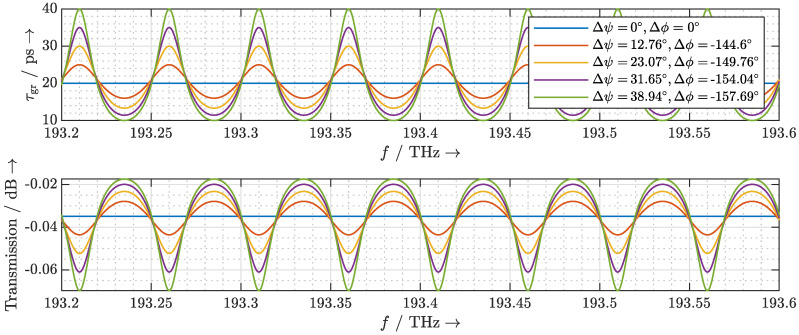
Calculated group delay response and transmission spectrum of an ORR with $f_{\mathrm{FSR},\mathrm{ORR}}=50\;\mathrm{GHz}$. The round-trip phase Δϕ is tuned so that for different values of Δψ the resonant frequencies of the ORR stay the same.

**Figure 5 sensors-20-05568-f005:**
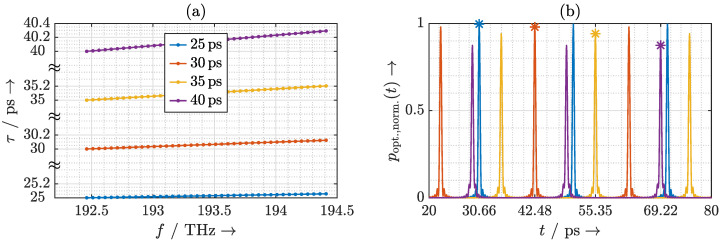
(**a**) Calculated delay τ that the ORR introduces to the laser modes and (**b**) the resulting instantaneous optical power $p_{\mathrm{opt}}\left(t\right)$ considering the frequency dependence of the phase shifters.

**Figure 6 sensors-20-05568-f006:**
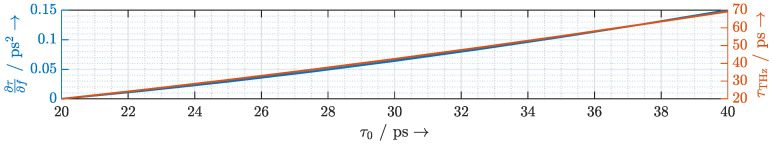
Slope $\frac{\partial \tau }{\partial f}$ of the delay that the ORR introduces to the laser modes (in blue) and the delay $\tau_{\mathrm{THz}}$ of the resulting terahertz pulse (in red) as a function of the delay $\tau_{0}$ that the ORR introduces to the first significant laser mode.

**Figure 7 sensors-20-05568-f007:**
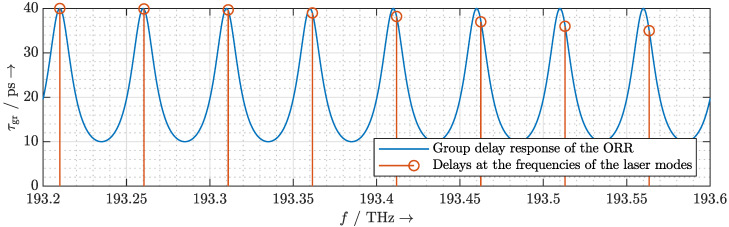
Calculated group delay response of the ORR (in blue) and the group delay at the laser modes of the monolithic mode-locked laser diode (MLLD) whose free spectral range (FSR) is slightly different from the FSR of the ORR (in red).

**Figure 8 sensors-20-05568-f008:**
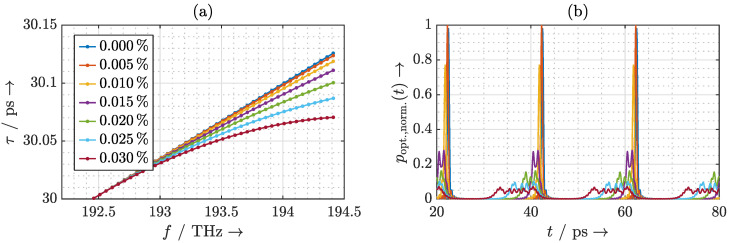
(**a**) Calculated delay τ that the ORR introduces to the laser modes and (**b**) the resulting instantaneous optical power $p_{\mathrm{opt}}\left(t\right)$ considering the frequency dependence of the phase shifters.

**Figure 9 sensors-20-05568-f009:**
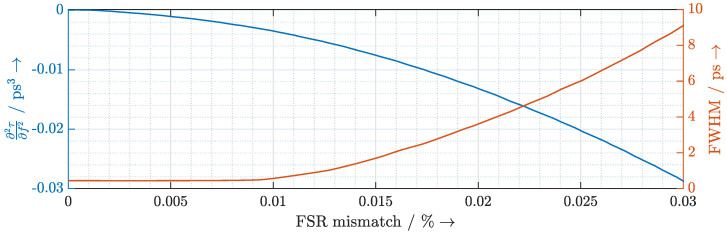
Calculated third-order dispersion $\frac{\partial^{2}\tau }{\partial f^{2}}$ that the ORR introduces to the laser modes (in blue) and the pulse width (full-width at half-maximum (FWHM)) of the resulting terahertz pulse (in red).

**Figure 10 sensors-20-05568-f010:**
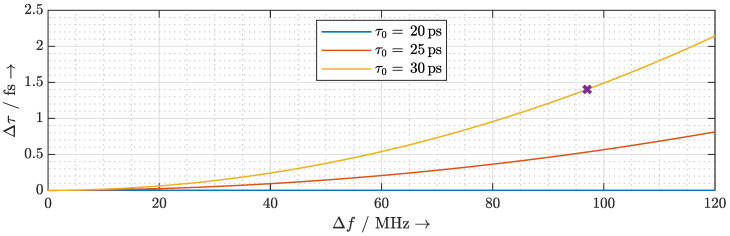
The delay fluctuation caused by the laser mode frequency fluctuation Δf.

**Figure 11 sensors-20-05568-f011:**
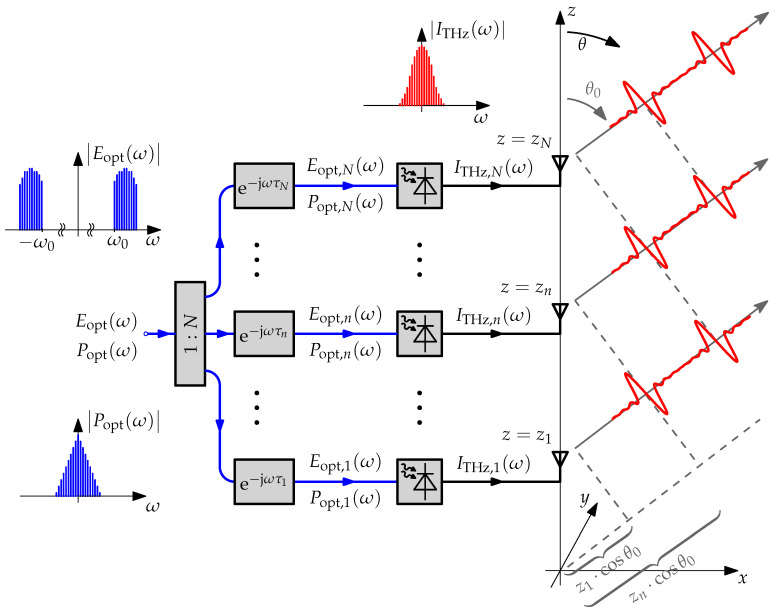
*N*-element linear antenna array with photonic quasi-true time delay beam steering network. The normalized optical fields are denoted by $E_{\mathrm{opt},n}\left(\omega \right)$, the normalized instantaneous optical powers are denoted by $P_{\mathrm{opt},n}\left(\omega \right)$, and the terahertz photocurrents generated by the photodiodes are denoted by $I_{\mathrm{THz},n}\left(\omega \right)$. The insets depict exemplary spectra for illustrative purposes. The time-domain pulses emitted by the antenna array are sketched for the case where the beam is steered to $\theta_{0}<90^{\circ }$.

**Figure 12 sensors-20-05568-f012:**
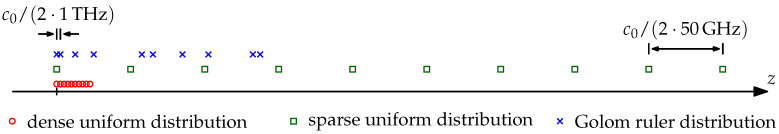
Three different element distributions for a linear 10-element array.

**Figure 13 sensors-20-05568-f013:**
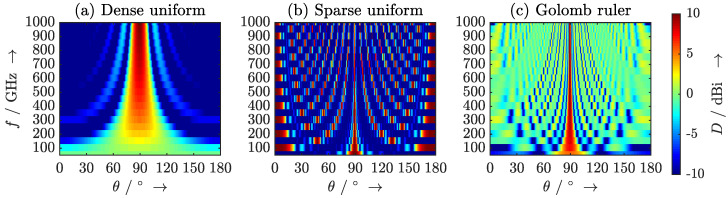
Calculated radiation patterns for the broadside case for (**a**) a dense uniform distribution, (**b**) a sparse uniform distribution, and (**c**) a distribution of the antenna elements according to a Golomb ruler.

**Figure 14 sensors-20-05568-f014:**
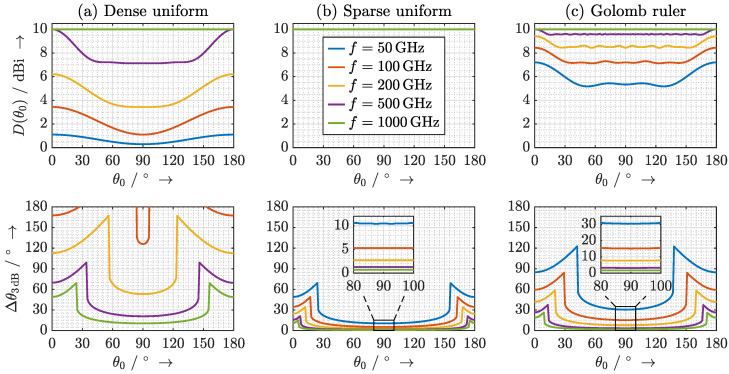
Calculated directivity (top row) and beamwidth (bottom row) for (**a**) a dense uniform distribution, (**b**) a sparse uniform distribution, and (**c**) a distribution of the antenna elements according to a Golomb ruler as a function of steering angle $\theta_{0}$ and operating frequency *f*. The directivity in the case of the sparse uniform array is identical to 10 dBi independent of steering angle and frequency.

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
