# Peer review of "Wideband Beam Steering Concept for Terahertz Time-Domain Spectroscopy: Theoretical Considerations"

_sensors, 2020, doi:10.3390/s20195568_

Round 1
Reviewer 1 Report
Authors are presenting a nice design idea for a relatively wideband operating, high repetition rate terahertz time-domain spectroscopy. The authors analytically and numerically show the possibility of using a linear array of emitters fed by a single source through a photonic beam steering network. The idea is novel and promising and would have a good impact for the scientific and industrial community.
Yet some statements are not clearly presented, I would recommend publication after a major revision.
I would like to thank all the Authors for their efforts and I kindly ask them to address my comments and suggestions below:
- The text is clearly written and well presented. The English language and style are fine still an overall check is required.
- I kindly ask the authors to add relevant references for all equations used.
- I suggest the Authors to better express in the text if the proposed design is only for a THz emitter or the authors assume to use the same method also for detection. If not, I kindly ask the authors to give further details on the detection methodology.
- The major advantage of THz-TDS is that the transient THz electric field is measured where the constituent elements of the pulse which are the amplitude and phase can be determined. I kindly ask the authors to comment on the phase characteristics of the proposed design. In the given design, the ORR is tuned by adjusting the phase difference between the arms. In accordance with the previous comment: If the same system is to be used for detection, the Authors must comment on the possible drawbacks when the system is to be used for material characterization since the signal amplitude and phase change due to the absorption and refraction processes in each sample.
- All calculations are given in the frequency range upto 1 THz (Fig 12, Fig 13 and Fig 14). In line 221 “Phase stability of 1_ up to 2 THz requires any delay fluctuations to be less than 1.4 fs”. I kindly ask the authors to extend their analytical analysis up to 2THz and/or show proof to this statement.
- Same as the comment 5: In Line 341-342 “the laser needs to be stabilized within 100MHz for the phase stability to be better than 1o up to 2 THz.”.
- Similarly: All calculations plot in figures 3 to 8 are given in around 190THz range. I kindly ask the authors to comment on the selection of the calculation band which is far from the design frequency (50GHz up to 1THz) and/or better explain the correlation if there is any. Else I kindly would suggest the authors to re-elaborate the given calculations within the design window.
- The figure caption in Figure 14. Is not clear, the differences in between the top row and bottom row plots must be better identified. More over there seems to be a typo in the x scale caption. Some of the plots must be re-scaled accordingly for a better visualization of the saturating lines.
Reviewer 2 Report
The manuscript reports on theoretical concepts of the terahertz time-domain spectroscopy. The scope of the paper is well demonstrated and achievements are interesting. I recommend to publish the manuscript in Sensors after authors consider items listed below.
1. I would acknowledge, if authors offered more information on the practical realization of the analyzed structure. For example, which material and fabrication technique should be used? Is the technology accessible in nowadays laboratories?
2. Fig. 10: The delay fluctuation for tau_0=20 ps looks zero. If it is really true, it should be stated explicitly. If it is nonzero, it should be properly visualized.
3. The right-hand side of Fig. 11, which shows the direction of the signal propagation, is improperly designed. Given scheme shows the propagation of the signal along the grating lobe and not in the main direction. Deducing from the figure description in the text, I guess that it is not an intention of this figure.
Reviewer 3 Report
The manuscript proposes an architecture for ultra-wideband beam steering in THz-TDS systems, based on properly designed optical ring resonators integrated with mode-locked lased diode. The analysis is sound and all the effect of the various building blocks is clearly simulated, including some study on the tolerance of the proposed system in terms of fabrication inaccuracies. Both the scientific quality and the presentation level are high. An experimental demonstration would raise any doubts on the feasibility and performance of the system, however it is understood that this is beyond the scope of this work. Publication of the manuscript in Sensors is recommended.
Reviewer 4 Report
The manuscript has described a so-called “Wideband Beam Steering Concept for Terahertz Time-Domain Spectroscopy”. Basically, the authors proposed to implement a method similar to time division multiplexing to pump each unit of THz antenna array by different pulses produced from an optical ring resonators. The far field THz pattern therefore could be controlled by modulating the phases among different THz pulse emitters. The “concept” is very straightforward. However, I could not recommend it to be published because it lacks of serious practical consideration. In addition, the theoretical analysis procedure and derivation process have already been well known and might be found extensively in literatures. In details:
- The method requires laser pulses running at very high repetition. Will the pumping laser provide sufficient energy, bandwidth or stability to pump the THz antenna array?
- How is the temporal splitting of the laser pulse train realized?
- How will the uniformity of the THz antenna array affect the performance?
- The dispersion control is a very delicate task. What is promising technique to ensure the required high precision?
- Assuming the proposed scheme would be an all fiber system, how will be nonlinear effect inside the transmission fibers?
In summary, the idea proposed by the authors was impractical at all so far.
Round 2
Reviewer 1 Report
I would like to thank all the Authors for their efforts.
The authors have answered my comments on the major issues satisfactorily. As a remaining minor issue, I would suggest authors to add relevant references for the “Equations (6), (7), (8), and (9) which are the fundamental equations” to help readers of diverse backgrounds. However it is only my opinion and I leave it to the discretion of the authors.
I would recommend publication in its present form.
With my best regards,
Reviewer 4 Report
The idea is good but not practical at all so far since many important practical details have been ignored. My personal feeling is that someone is proposing ‘I want to build a spacecraft to land on Sun. But if there is any material can sustained the high temperature of Sun is out of my scope.‘
